# An algorithm for $\ell_1$ nearest neighbor search via monotonic embedding

**Xinan Wang**[*]
UC San Diego
xinan@ucsd.edu

**Sanjoy Dasgupta**
UC San Diego
dasgupta@cs.ucsd.edu

## Abstract

Fast algorithms for nearest neighbor (NN) search have in large part focused on $\ell_2$ distance. Here we develop an approach for $\ell_1$ distance that begins with an explicit and exactly distance-preserving embedding of the points into $\ell_2^2$. We show how this can efficiently be combined with random-projection based methods for $\ell_2$ NN search, such as locality-sensitive hashing (LSH) or random projection trees. We rigorously establish the correctness of the methodology and show by experimentation using LSH that it is competitive in practice with available alternatives.

## 1    Introduction

Nearest neighbor (NN) search is a basic primitive of machine learning and statistics. Its utility in practice hinges on two critical issues: (1) picking the right distance function and (2) using algorithms that find the nearest neighbor, or an approximation thereof, quickly.

The default distance function is very often Euclidean distance. This is a matter of convenience and can be partially justified by theory: a classical result of Stone [1] shows that $k$-nearest neighbor classification is universally consistent in Euclidean space. This means that no matter what the distribution of data and labels might be, as the number of samples $n$ goes to infinity, the $k_n$-NN classifier converges to the Bayes-optimal decision boundary, for any sequence $(k_n)$ with $k_n \to \infty$ and $k_n/n \to 0$. The downside is that the *rate* of convergence could be slow, leading to poor performance on finite data sets. A more careful choice of distance function can help, by better separating the different classes. For the well-known MNIST data set of handwritten digits, for instance, the 1-NN classifier using Euclidean distance has an error rate of about 3%, whereas a more careful choice of distance function—tangent distance [2] or shape context [3], for instance—brings this below 1%.

The second impediment to nearest neighbor search in practice is that a naive search through $n$ candidate neighbors takes $O(n)$ time, ignoring the dependence on dimension. A wide variety of ingenious data structures have been developed to speed this up. The most popular of these fall into two categories: *hashing-based* and *tree-based*.

Perhaps the best-known hashing approach is *locality-sensitive hashing* (LSH) [4, 5, 6, 7, 8, 9, 10]. These randomized data structures find approximate nearest neighbors with high probability, where $c$-approximate solutions are those that are at most $c$ times as far as the nearest neighbor.

Whereas hashing methods create a lattice-like spatial partition, tree methods [11, 12, 13, 14] create a hierarchical partition that can also be used to speed up nearest neighbor search. There are families of randomized trees with strong guarantees on the tradeoff between query time and probability of finding the exact nearest neighbor [15].

These hashing and tree methods for $\ell_2$ distance both use the same primitive: *random projection* [16]. For data in $\mathbb{R}^d$, they (repeatedly) choose a random direction $u$ from the multivariate Gaussian

---

[*]Supported by UC San Diego Jacobs Fellowship

$N(0, I_d)$ and then project points $x$ onto this direction: $x \mapsto u \cdot x$. Such projections have many appealing mathematical properties that make it possible to give algorithmic guarantees, and that also produce good performance in practice.

For distance functions other than $\ell_2$, there has been far less work. In this paper, we develop nearest neighbor methods for $\ell_1$ distance. This is a more natural choice than $\ell_2$ in many situations, for instance when the data points are probability distributions: documents are often represented as distributions over topics, images as distributions over categories, and so on. Earlier works on $\ell_1$ search are summarized below. We adopt a different approach, based on a novel embedding.

One basic fact is that $\ell_1$ distance is not embeddable in $\ell_2$ [17]. That is, given a set of points $x_1, \ldots, x_n \in \mathbb{R}^d$, it is in general not possible to find corresponding points $z_1, \ldots, z_n \in \mathbb{R}^q$ such that $\|x_i - x_j\|_1 = \|z_i - z_j\|_2$. This can be seen even from the four points at the vertices of a square—any embedding of these into $\ell_2$ induces a multiplicative distortion of at least $\sqrt{2}$.

Interestingly, however, the *square root* of $\ell_1$ distance *is* embeddable in $\ell_2$ [18]. And the nearest neighbor with respect to $\ell_1$ distance is the same as the nearest neighbor with respect to $\ell_1^{1/2}$. This observation is the starting point of our approach. It suggests that we might be able to embed data into $\ell_2$ and then simply apply well-established methods for $\ell_2$ nearest neighbor search. However, there are numerous hurdles to overcome.

First, the embeddability of $\ell_1^{1/2}$ into $\ell_2$ is an existential, not algorithmic, fact. Indeed, all that is known for general case is that there exists such an embedding into Hilbert space. For the special case of data in $\{0, 1, \ldots, M\}^d$, earlier work has suggested a unary embedding into a Hamming space $\{0, 1\}^{Md}$ (where $0 \le x \le M$ gets mapped to $x$ 1's followed by $(M - x)$ 0's) [19], but this is wasteful of space and is inefficient to be used by dimension reduction algorithms [16] when $M$ is large. Our embedding is general and is more efficient.

Now, given a finite point set $x_1, \ldots, x_n \in \mathbb{R}^d$ and the knowledge that an embedding exists, we could use multidimensional scaling [20] to find such an embedding. But this takes $O(n^3)$ time, which is often not viable. Instead, we exhibit an *explicit* embedding: we give an expression for points $z_1, \ldots, z_n \in \mathbb{R}^{O(nd)}$ such that $\|x_i - x_j\|_1 = \|z_i - z_j\|_2^2$.

This brings us to the second hurdle. The explicit construction avoids infinite-dimensional space but is still much higher-dimensional than we would like. The space requirement for writing down the $n$ embedded points is $O(n^2 d)$, which is prohibitive in practice. To deal with this, we recall that the two popular schemes for $\ell_2$ embedding described above are both based on Gaussian random projections, and in fact look at the data *only* through the lens of such projections. We show how to compute these projections without ever constructing the $O(nd)$-dimensional embeddings explicitly.

Finally, even if it is possible to efficiently build a data structure on the $n$ points, how can queries be incorporated? It turns out that if a query point is added to the original $n$ points, our explicit embedding changes significantly. Nonetheless, by again exploiting properties of Gaussian random projections, we show that it is possible to hold on to the random projections of the original $n$ embedded points and to set the projection of the query point so that the correct joint distribution is achieved. Moreover, this can be done very efficiently.

Finally, we run a variety of experiments showing the good practical performance of this approach.

### Related work

The $k$-d tree [11] is perhaps the prototypical tree-based method for nearest neighbor search, and can be used for $\ell_1$ distance. It builds a hierarchical partition of the data using coordinate-wise splits, and uses geometric reasoning to discard subtrees during NN search. Its query time can degenerate badly with increasing dimension, as a result of which several variants have been developed, such as trees in which the cells are allowed to overlap slightly [21]. Various tree-based methods have also been developed for general metrics, such as the metric tree and cover tree [14, 12].

For $k$-d tree variants, theoretical guarantees are available for exact $\ell_2$ nearest neighbor search when the split direction is chosen at random from a multivariate Gaussian [15]. For a data set of $n$ points, the tree has size $O(n)$ and the query time is $O(2^d \log n)$, where $d$ is the intrinsic dimension of the data. Such analysis is not available for $\ell_1$ distance.

Also in wide use is locality-sensitive hashing for approximate nearest neighbor search [22]. For a data set of $n$ points, this scheme builds a data structure of size $O(n^{1+\rho})$ and finds a $c$-approximate nearest neighbor in time $O(n^\rho)$, for some $\rho > 0$ that depends on $c$, on the specific distance function, and on the hash family. For $\ell_2$ distance, it is known how to achieve $\rho \approx 1/c^2$ [23], although the scheme most commonly used in practice has $\rho \approx 1/c$ [8]. This works by repeatedly using the following hash function:

$$ h(x) \;=\; \lfloor (v \cdot x + b)/R \rfloor, $$

where $v$ is chosen at random from a multivariate Gaussian, $R > 0$ is a constant, and $b$ is uniformly distributed in $[0, R)$. A similar scheme also works for $\ell_1$, using Cauchy random projection: each coordinate of $v$ is picked independently from a standard Cauchy distribution. This achieves exponent $\rho \approx 1/c$, although one downside is the high variance of this distribution. Another LSH family [22, 10] uses a randomly shifted grid for $\ell_1$ nearest neighbor search. But it is less used in practice, due to its restrictions on data. For example, if the nearest neighbor is further away than the width of the grid, it may never be found.

Besides LSH, random projection is the basis for some other NN search algorithms [24, 25], classification methods [26], and dimension reduction techniques [27, 28, 29].

There are several impediments to developing NN methods for $\ell_1$ spaces. 1) There is no Johnson-Lindenstrauss type dimension reduction technique for $\ell_1$ [30]. 2) The Cauchy random projection does not preserve the $\ell_1$ distance as a norm, which restricts its usage for norm based algorithms [31]. 3) Useful random properties [26] cannot be formulated exactly; only approximations exist. Fortunately, all these three problems are absent in $\ell_2$ space, which motivates developing efficient embedding algorithms from $\ell_1$ to $\ell_2$.

## 2 Explicit embedding

We begin with an explicit isometric embedding from $\ell_1$ to $\ell_2^2$ for 1-dimensional data. This extends immediately to multiple dimensions because both $\ell_1$ and $\ell_2^2$ distance are coordinatewise additive.

### 2.1 The 1-dimensional case

First, sort the points $x_1, \ldots, x_n \in \mathbb{R}$ so that $x_1 \leq x_2 \leq \cdots \leq x_n$. Then, construct the embedding $\phi(x_1), \phi(x_2), \ldots, \phi(x_n) \in \mathbb{R}^{n-1}$ as follows:

$$
\overbrace{\begin{bmatrix} 0 \\ 0 \\ 0 \\ \vdots \\ 0 \end{bmatrix}}^{\phi(x_1)}
\overbrace{\begin{bmatrix} \sqrt{x_2 - x_1} \\ 0 \\ 0 \\ \vdots \\ 0 \end{bmatrix}}^{\phi(x_2)}
\overbrace{\begin{bmatrix} \sqrt{x_2 - x_1} \\ \sqrt{x_3 - x_2} \\ 0 \\ \vdots \\ 0 \end{bmatrix}}^{\phi(x_3)}
\cdots
\overbrace{\begin{bmatrix} \sqrt{x_2 - x_1} \\ \sqrt{x_3 - x_2} \\ \sqrt{x_4 - x_3} \\ \vdots \\ \sqrt{x_n - x_{n-1}} \end{bmatrix}}^{\phi(x_n)}
\tag{1}
$$

For any $1 \leq i < j \leq n$, $\phi(x_i)$ and $\phi(x_j)$ agree on all coordinates except $i$ to $(j-1)$. Therefore,

$$ \|\phi(x_i) - \phi(x_j)\|_2 = \left[ \sum_{k=i+1}^{j} \left( \sqrt{x_k - x_{k-1}} \right)^2 \right]^{1/2} = \left[ \sum_{k=i+1}^{j} x_k - x_{k-1} \right]^{1/2} = |x_j - x_i|^{1/2}, \tag{2} $$

so the embedding preserves the $\ell_1^{1/2}$ distance between these points. Since the construction places no restrictions on the range of $x_1, x_2, \ldots, x_n$, it is applicable to any finite set of points.

### 2.2 Extension to multiple dimensions

We construct an embedding of $d$-dimensional points by stacking 1-dimensional embeddings.

Consider points $x_1, x_2, \ldots, x_n \in \mathbb{R}^d$. Suppose we have a collection of embedding maps $\phi_1, \phi_2, \ldots, \phi_d$, one per dimension. Each of the embeddings is constructed from the values on a single coordinate: if we let $x_i^{(j)}$ denote the $j$-th coordinate of $x_i$, for $1 \leq j \leq d$, then embedding $\phi_j$

is based on $x_1^{(j)}, x_2^{(j)}, \ldots, x_n^{(j)} \in \mathbb{R}$. The overall embedding is the concatenation

$$\phi(x_i) = \left( \phi_1^\tau \left( x_i^{(1)} \right), \phi_2^\tau \left( x_i^{(2)} \right), \ldots, \phi_d^\tau \left( x_i^{(d)} \right) \right)^\tau \in \mathbb{R}^{d(n-1)} \tag{3}$$

where $1 \le i \le n$, and $\tau$ denotes transpose. For any $1 \le i < j \le n$,

$$\|\phi(x_i) - \phi(x_j)\|_2 = \left[ \sum_{k=1}^d \left\| \phi_k \left( x_i^{(k)} \right) - \phi_k \left( x_j^{(k)} \right) \right\|_2^2 \right]^{1/2} \tag{4}$$

$$= \left[ \sum_{k=1}^d \left| x_i^{(k)} - x_j^{(k)} \right| \right]^{1/2} = \|x_i - x_j\|_1^{1/2} \tag{5}$$

It may be of independent interest to consider the properties of this explicit embedding. We can represent it by a matrix of $n$ columns with one embedded point per column. The rank of this matrix—and, therefore, the dimensionality of the embedded points—turns out to be $O(n)$. But we can show that the "effective rank" [32] of the centered matrix is just $O(d \log n)$; see Appendix B.

## 3 Incorporating a query

Once again, we begin with the 1-dimensional case and then extend to higher dimension.

### 3.1 The 1-dimensional case

For nearest neighbor search, we need a joint embedding of the data points $\mathcal{S} = \{x_1, x_2, \ldots, x_n\}$ with the subsequent query point $q$. In fact, we need to embed $\mathcal{S}$ first and then incorporate $q$ later, but this is non-trivial since adding $q$ changes the explicit embedding of other points.

We start with an example. Again, assume $x_1 \le x_2 \le \cdots \le x_n$.

**Example 1.** *Suppose query $q$ has $x_2 \le q < x_3$. Adding $q$ to the original $n$ points changes the embedding $\phi(\cdot) \in \mathbb{R}^{n-1}$ of Eq. 1 to $\overline{\phi}(\cdot) \in \mathbb{R}^n$. Notice that the dimension increases by one.*

$$
\overbrace{\begin{bmatrix} 0 \\ 0 \\ 0 \\ 0 \\ \vdots \\ 0 \end{bmatrix}}^{\overline{\phi}(x_1)}
\overbrace{\begin{bmatrix} \sqrt{x_2 - x_1} \\ 0 \\ 0 \\ 0 \\ \vdots \\ 0 \end{bmatrix}}^{\overline{\phi}(x_2)}
\overbrace{\begin{bmatrix} \sqrt{x_2 - x_1} \\ \sqrt{q - x_2} \\ \sqrt{x_3 - q} \\ 0 \\ \vdots \\ 0 \end{bmatrix}}^{\overline{\phi}(x_3)}
\cdots
\overbrace{\begin{bmatrix} \sqrt{x_2 - x_1} \\ \sqrt{q - x_2} \\ \sqrt{x_3 - q} \\ \sqrt{x_4 - x_3} \\ \vdots \\ \sqrt{x_n - x_{n-1}} \end{bmatrix}}^{\overline{\phi}(x_n)}
\tag{6}
$$

*The query point is mapped to $\overline{\phi}(q) = (\sqrt{x_2 - x_1}, \sqrt{q - x_2}, 0, \ldots, 0)^\tau \in \mathbb{R}^n$.*

From the example above, it is clear what happens when $q$ lies between some $x_i$ and $x_{i+1}$. There are also two "corner cases" that can occur: $q < x_1$ and $q > x_n$. Fortunately, the embedding of $\mathcal{S}$ is almost unchanged for the corner cases: $\overline{\phi}(x_i) = (\phi^\tau(x_i), 0)^\tau \in \mathbb{R}^n$, appending a zero at the end. For $q < x_1$, the query is mapped to $\overline{\phi}(q) = (0, \ldots, 0, \sqrt{x_1 - q})^\tau \in \mathbb{R}^n$; for $q \ge x_n$, the query is mapped to $\overline{\phi}(q) = (\sqrt{x_2 - x_1}, \sqrt{x_3 - x_2}, \ldots, \sqrt{x_n - x_{n-1}}, \sqrt{q - x_n})^\tau \in \mathbb{R}^n$.

### 3.2 Random projection for the 1-dimensional case

We would like to generate Gaussian random projections of the $\ell_2$ embeddings of the data points. In this subsection, we mainly focus on the typical case when the query $q$ lies between two data points, and we leave the treatment of the (simpler) corner cases to Alg. 1. The notation follows section 3.1, and we assume the $x_i$ are arranged in increasing order for $i = 1, 2, \ldots, n$.

**Setting 1.** *The query lies between two data points: $x_\alpha \le q < x_{\alpha+1}$ for some $1 \le \alpha \le n - 1$.*

We will consider two methods for randomly projecting the embedding of $\mathcal{S} \cup \{q\}$ and show that they yield exactly the same joint distribution.

The first method applies Gaussian random projection to the embedding $\overline{\phi}$ of $\mathcal{S} \cup \{q\}$. Sample a multivariate Gaussian vector $v$ from $N(0, I_n)$. For any $x \in \mathcal{S} \cup \{q\}$, the projection is

$$p_g(x) := v \cdot \overline{\phi}(x) \tag{7}$$

This is exactly the projection we want. However, it requires both $\mathcal{S}$ and $q$, whereas in practice, we will initially have to project just $\mathcal{S}$ by itself, and we will only later be given some (arbitrary) $q$.

The second method starts by projecting the explicitly embedded points $\mathcal{S}$. Later, it receives query $q$ and finds a suitable projection for it as well. So, we begin by sampling a multivariate Gaussian vector $u$ from $N(0, I_{n-1})$, and for any $x \in \mathcal{S}$, use the projection

$$p_e(x) := u \cdot \phi(x) \tag{8}$$

where the subindex $e$ stands for embedding. Conditioned on the value $(p_e(x_{\alpha+1}) - p_e(x_\alpha))$, namely $\sqrt{x_{\alpha+1} - x_\alpha} \cdot u^{(\alpha)}$, the projection of a subsequent query $q$ is taken to be

$$p_e(q) = p_e(x_\alpha) + \Delta$$
$$\Delta \sim \mathcal{N}\left( \frac{\sigma_1^2 \cdot (p_e(x_{\alpha+1}) - p_e(x_\alpha))}{\sigma_1^2 + \sigma_2^2}, \frac{\sigma_1^2 \sigma_2^2}{\sigma_1^2 + \sigma_2^2} \right) \tag{9}$$

where $\sigma_1^2 = q - x_\alpha$, $\sigma_2^2 = x_{\alpha+1} - q$.

**Theorem 1.** *Fix any $x_1, \ldots, x_n, q \in \mathbb{R}$ satisfying Setting 1. Consider the joint distribution of $[p_g(x_1), p_g(x_2), \ldots, p_g(x_n), p_g(q)]$ induced by a random choice of $v$ (as per Eq. 7), and the joint distribution of $[p_e(x_1), p_e(x_2), \ldots, p_e(x_n), p_e(q)]$ induced by a random choice of $u$ and $\Delta$ (as per Eqs. 8 and 9). These distributions are identical.*

The details are in Appendix A: briefly, we show that both joint distributions are multivariate Gaussians, and that they have the same mean and covariance.

We highlight the advantages of our method. First, projecting the data set using Eq. 8 does not require advance knowledge of the query, which is crucial for nearest neighbor search; second, generating the projection for the 1-dimensional query takes $O(\log n)$ time, which makes this method efficient. We describe the 1-dimensional algorithm in Alg. 1, where we assume that a permutation that sorts the points, denoted $\Pi$, is provided, along with the location of $q$ within this ordering, denoted $\alpha$. We will resolve this later in Alg. 2.

### 3.3 Random projection for the higher dimensional case

We will henceforth use ERP (Euclidean random projection) to denote our overall scheme consisting of embedding $\ell_1$ into $\ell_2^2$, followed by random Gaussian projection (Alg. 2). A competitor scheme, as described earlier, applies Cauchy random projection directly in the $\ell_1$ space; we refer to this as CRP. The time and space costs for ERP are shown in Table 1, if we generate $k$ projections for $n$ data points and $m$ queries in $\mathbb{R}^d$. The costs scale linearly in $d$, since the constructions and computation are dimension by dimension. We have a detailed analysis below.

*Preprocessing:* This involves sorting the points along each coordinate separately and storing the resulting permutations $\Pi_1, \ldots, \Pi_d$. The time and space costs are acceptable, because reading or storing the data takes as much as $O(nd)$.

*Project data:* The time taken by ERP to project the $n$ points is comparable to that of CRP. But ERP requires a factor $O(n)$ more space, compared to $O(kd)$ for CRP, because it needs to store the projections of each of the individual coordinates of the data points.

*Project query:* ERP methods are efficient for query answering. The projection is calculated directly in the original $d$-dimensional space. The $\log n$ overhead comes from using binary search, coordinatewise, to place the query within the ordering of the data points. Once these ranks are obtained, they can be reused for as many projections as needed.

---
**Algorithm 1** Random projection (1-dimensional case)
---

**function** project-data $(\mathcal{S}, \Pi)$
**input:**
   — data set $\mathcal{S} = (x_i : 1 \le i \le n)$
   — sorted indices $\Pi = (\pi_i : 1 \le i \le n)$
     such that $x_{\pi_1} \le x_{\pi_2} \le, \ldots, \le x_{\pi_n}$
**output:**
   — projections $P = (p_i : 1 \le i \le n)$ for $\mathcal{S}$

$p_{\pi_1} \leftarrow 0$
**for** $i = 2, 3, \ldots, n$ **do**
$u_i \leftarrow \mathcal{N}(0, 1)$
$p_{\pi_i} \leftarrow p_{\pi_{i-1}} + u_i \cdot \sqrt{x_{\pi_i} - x_{\pi_{i-1}}}$
**end for**
**return** $P$

**function** project-query$(q, \alpha, \mathcal{S}, \Pi, P)$
**input:**
   — query $q$ and its rank $\alpha$ in data set $\mathcal{S}$
   — sorted indices $\Pi$ of $\mathcal{S}$
   — projections $P$ of $\mathcal{S}$
**output:**
   — projection $p_q$ for $q$

**case:** $1 \le \alpha \le n - 1$
$\sigma_1^2 \leftarrow q - x_{\pi_\alpha}$
$\sigma_2^2 \leftarrow x_{\pi_{\alpha+1}} - q$
$\Delta \leftarrow \mathcal{N} \left( \dfrac{\sigma_1^2 \cdot (p_{\pi_{\alpha+1}} - p_{\pi_\alpha})}{\sigma_1^2 + \sigma_2^2}, \dfrac{\sigma_1^2 \sigma_2^2}{\sigma_1^2 + \sigma_2^2} \right)$
$p_q \leftarrow p_{\pi_\alpha} + \Delta$
**case:** $\alpha = 0$
   $r \leftarrow \mathcal{N}(0, 1), \quad p_q \leftarrow r \cdot \sqrt{x_{\pi_1} - q}$
**case:** $\alpha = n$
   $r \leftarrow \mathcal{N}(0, 1), \quad p_q \leftarrow p_{\pi_n} + r \cdot \sqrt{q - x_{\pi_n}}$
**return** $p_q$

---

Table 1: Efficiency of ERP algorithm: Generate $k$ projections for $n$ data points and $m$ queries in $\mathbb{R}^d$.

|  | Preprocessing | Project data | Project query |
|---|---|---|---|
| Time cost | $O(dn \log n)$ | $O(knd)$ | $O(md(k + \log n))$ |
| Space cost | $O(dn)$ | $O(knd)$ | NA |

## 4 Experiment

In this section, we demonstrate that ERP can be directly used by existing NN search algorithms, such as LSH, for efficient $\ell_1$ NN search. We choose commonly used data sets for image retrieval and text classification. Besides our method, we also implement the metric tree (a popular tree-type data structure) and Cauchy LSH for comparison.

**Data sets** When data points represent distributions, $\ell_1$ distance is natural. We use four such data sets. 1) Corel_uci [21], available at [33], contains 68,040 histograms (32-dimension) for color images from Corel image collections; 2) Corel_hist [34, 21], processed by [21], contains 19,797 histograms (64-dimension, non-zero dimension is 44) for color images from Corel Stock Library; 3) Cade [35], is a collection of documents from Brazilian web pages. Topics are extracted using latent Dirichlet allocation algorithm [36]. We use 13,016 documents with distributions over the 120 topics (120-dimension); 4) We download about 35,000 images from ImageNet [37], and process each of them into a probabilistic distribution over 1,000 classes using trained convolution neural network [38]. Furthermore, we collapse the distribution into a 100-dimension representation, summing each 10 consecutive mass of probability. This reduces the training and testing time.

In each data set, we remove duplicates. For either parameter optimization or testing, we randomly separate out $10\%$ of the data as queries such that the query-to-data ratio is $1 : 9$.

**Performance evaluation** We evaluate performance using query cost. For linear scan or metric tree, this is the average number of points accessed when answering a query. For LSH, we also need to add the overhead of evaluating the LSH functions.

The scheme [8, 39] of LSH is summarized as follows. Given three parameters $k$, $L$ and $R$ ($k, L$ are positive integers, $k$ is even, $R$ is a positive real), the LSH algorithm uses $k$-tuple hash functions of the form $g(x) = (h_1(x), h_2(x), \ldots, h_k(x))$ to distribute data or queries to their bins. $L$ is the total number of such $g$-functions. The $h$-functions are of the form $h(x) = \lfloor (v \cdot x + b)/R \rfloor$, each

---
**Algorithm 2** Overall algorithm for Random projection, in context of NN search
---

Starting information:
— data set $\mathcal{S} = \{x_i : 1 \leq i \leq n\} \subset \mathbb{R}^d$
Subsequent arrival:
— query $q \in \mathbb{R}^d$

**preprocessing:**
Sort data along each dimension:
**for** $j \in \{1, \ldots, d\}$ **do**
$\quad \mathcal{S}_j = \{x_i^{(j)} : 1 \leq i \leq n\}$
$\quad \Pi_j \leftarrow$ index-sort $(\mathcal{S}_j)$, where
$\quad\quad \Pi_j = \{\pi_{ji} : 1 \leq i \leq n\}$ satisfying
$\quad\quad x_{\pi_{j1}}^{(j)} \leq x_{\pi_{j2}}^{(j)} \leq \cdots \leq x_{\pi_{jn}}^{(j)}$
**end for**
**save** $\Pi = (\Pi_1, \Pi_2, \ldots, \Pi_d)$

**project data:**
**for** $j = 1, 2, \ldots, d$ **do**
$\quad P_j \leftarrow$ project-data $(\mathcal{S}_j, \Pi_j)$ where
$\quad\quad P_j = \{p_{ji} : 1 \leq i \leq n\}$
**end for**
**save** $P = (P_1, P_2, \ldots, P_d)$
projection of $x_i \in \mathcal{S}$ is $\sum_{j=1}^d p_{ji}$

**project query:**
**for** $j = 1, 2, \ldots, d$ **do**
$\quad \alpha_j \leftarrow$ binary-search$(q^{(j)}, S_j, \Pi_j)$ satisfying
$\quad\quad x_{\pi_{j\alpha_j}}^{(j)} \leq q^{(j)} \leq x_{\pi_{j(\alpha_j+1)}}^{(j)}$
**end for**
**save** rank $\alpha$ for use in multiple projections
$p_q \leftarrow 0$
**for** $j = 1, 2, \ldots, d$ **do**
$\quad p_g \leftarrow p_g +$ project-query$(q^{(j)}, \alpha_j, \mathcal{S}_j, \Pi_j, P_j)$
**end if**
projection for $q$ is $p_g$

Table 2: Performance evaluation: Query cost = $T_r + T_o$.

|  | Retrieval cost: $T_r$ | Overhead: $T_o$ |
|---|---|---|
| Linear Scan or Metric Tree | # Accessed points | 0 |
| CRP-LSH | # Accessed points | $k/2 \cdot \sqrt{2L}$ |
| ERP-LSH | # Accessed points | $k/2 \cdot \sqrt{2L} + \log n$ |

either explicitly or implicitly associated with a random vector $v$ and a uniformly distributed variable $b \in [0, R)$. As suggested in [39], we implement the reuse of $h$-functions so that only $(k/2 \cdot \sqrt{2L})$ of them are actually evaluated. For ERP-LSH, there is an additional overhead of $\log n$ due to the use of binary search. We summarize these costs in Table 2; for conciseness, we have removed the linear dependence on $d$ in both the retrieval cost and the overhead.

**Implementations** The linear scan and the metric tree are for exact NN search. We use the code [40] for metric tree. For LSH, there is only public code for $\ell_2$ NN search. We implement the LSH scheme, referring to the manual [39]. In particular, we implement the reuse of the $h$-functions, such that the number of actually evaluated $h$-functions is $(k/2 \cdot \sqrt{2L})$, in contrast to $(k \cdot L)$.

We choose approximation factor $c = 1.5$ (the results turn out to be much closer to true NN), and set the success rate to be $0.9$, which means that the algorithm should report $c$-approximate NN successfully for at least $90\%$ of the queries. Taking the parameter suggestions [8] into account, we choose $R$ for CRP-LSH from $d_{NN} \times \{1, 5, 10, 50, 100\}$; we choose $R$ for ERP-LSH from $d'_{NN} \times \{1, 2, 3, 4\}$, where $d_{NN} = \frac{1}{|Q|} \sum_{q \in Q} \|q - x_{NN}(q)\|_1$ is the average $\ell_1$ NN distance; $d'_{NN} = \frac{1}{|Q|} \sum_{q \in Q} \sqrt{\|q - x_{NN}(q)\|_1}$ is the average $\ell_1^{1/2}$ NN distance. The term $d_{NN}$ or $d'_{NN}$ normalizes the average NN distance to 1 for LSH. Fixing $R$, we optimize $k$ and $L$ in the following range: $k \in \{2, 4, \ldots, 30\}$, $L \in \{1, 2, \ldots, 40\}$.

**Results** Both CRP-LSH and ERP-LSH achieve a competitive efficiency over the other two methods. We list the test results in Table 3, and put parameters in Table 4 in Appendix C.

Table 3: Average query cost and average approximation rate if applicable (in parentheses).

| | Corel_uci $(d = 32)$ | Corel_hist $(d = 44)$ | Cade $(d = 120)$ | ImageNet $(d = 100)$ |
|---|---|---|---|---|
| Linear scan | 61220 | 17809 | 11715 | 31458 |
| Metric tree | 2575 | 718 | 9184 | 12375 |
| CRP-LSH | $329 \pm 55$ (1.07) | $245 \pm 43$ (1.05) | $292 \pm 11$ (1.11) | $548 \pm 66$ (1.09) |
| ERP-LSH | $330 \pm 18$ (1.11) | $250 \pm 15$ (1.08) | $218 \pm 8$ (1.15) | $346 \pm 15$ (1.13) |

## 5  Conclusion

In this paper, we have proposed an explicit embedding from $\ell_1$ to $\ell_2^2$, and we have found an algorithm to generate the random projections, reducing the time dependence of $n$ from $O(n)$ to $O(\log n)$. In addition, we have observed that the effective rank of the (centered) embedding is as low as $O(d \ln n)$, compared to its rank $O(n)$. Algorithms remain to be explored, in order to take advantage of such a low rank.

Our current method takes space $O(ndm)$ to store the parameters of the random vectors, where $m$ is the number of hash functions. We have implemented one empirical scheme [39] to reuse the hashing functions. It is still expected to develop other possible schemes.

## Acknowledgement

The authors are grateful to the National Science Foundation for support under grant IIS-1162581.

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
