[Supplementary Material]

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

   $\Pi_j = \{\pi_{ji} : \ 1 \le i \le n\}$ satisfying
   $x_{\pi_{j1}}^{(j)} \le x_{\pi_{j2}}^{(j)} \le \cdots \le x_{\pi_{jn}}^{(j)}$
 **end for**
 **save** $\Pi = (\Pi_1, \Pi_2, \ldots, \Pi_d)$

**project data:**
 **for** $j = 1, 2, \ldots, d$ **do**
  $P_j \leftarrow$ project-data $(\mathcal{S}_j, \Pi_j)$ where
   $P_j = \{p_{ji} : \ 1 \le i \le n\}$
 **end for**
 **save** $P = (P_1, P_2, \ldots, P_d)$
 projection of $x_i \in \mathcal{S}$ is $\sum_{j=1}^{d} p_{ji}$

**project query:**
 **for** $j = 1, 2, \ldots, d$ **do**
  $\alpha_j \leftarrow$ binary-search$(q^{(j)}, S_j, \Pi_j)$ satisfying
   $x_{\pi_{j\alpha_j}}^{(j)} \le q^{(j)} \le x_{\pi_{j(\alpha_j+1)}}^{(j)}$
 **end for**
 **save** rank $\alpha$ for use in multiple projections
 $p_q \leftarrow 0$
 **for** $j = 1, 2, \ldots, d$ **do**
 $p_g \leftarrow p_g +$ project-query$(q^{(j)}, \alpha_j, \mathcal{S}_j, \Pi_j, P_j)$
 **end if**
 projection for $q$ is $p_g$

Table 2: Performance evaluation: Query cost $= T_r + T_o$.

|  | Retrieval cost: $T_r$ | Overhead: $T_o$ |
|---|---|---|
| Linear Scan or Metric Tree | # Accessed points | 0 |
| CRP-LSH | # Accessed points | $k/2 \cdot \sqrt{2L}$ |
| ERP-LSH | # Accessed points | $k/2 \cdot \sqrt{2L} + \log n$ |

either explicitly or implicitly associated with a random vector $v$ and a uniformly distributed variable $b \in [0, R)$. As suggested in [39], we implement the reuse of $h$-functions so that only $(k/2 \cdot \sqrt{2L})$ of them are actually evaluated. For ERP-LSH, there is an additional overhead of $\log n$ due to the use of binary search. We summarize these costs in Table 2; for conciseness, we have removed the linear dependence on $d$ in both the retrieval cost and the overhead.

**Implementations**  The linear scan and the metric tree are for exact NN search. We use the code [40] for metric tree. For LSH, there is only public code for $\ell_2$ NN search. We implement the LSH scheme, referring to the manual [39]. In particular, we implement the reuse of the $h$-functions, such that the number of actually evaluated $h$-functions is $(k/2 \cdot \sqrt{2L})$, in contrast to $(k \cdot L)$.

We choose approximation factor $c = 1.5$ (the results turn out to be much closer to true NN), and set the success rate to be 0.9, which means that the algorithm should report $c$-approximate NN successfully for at least 90% of the queries. Taking the parameter suggestions [8] into account, we choose $R$ for CRP-LSH from $d_{NN} \times \{1, 5, 10, 50, 100\}$; we choose $R$ for ERP-LSH from $d'_{NN} \times \{1, 2, 3, 4\}$, where $d_{NN} = \frac{1}{|Q|} \sum_{q \in Q} \|q - x_{NN}(q)\|_1$ is the average $\ell_1$ NN distance; $d'_{NN} = \frac{1}{|Q|} \sum_{q \in Q} \sqrt{\|q - x_{NN}(q)\|_1}$ is the average $\ell_1^{1/2}$ NN distance. The term $d_{NN}$ or $d'_{NN}$ normalizes the average NN distance to 1 for LSH. Fixing $R$, we optimize $k$ and $L$ in the following range: $k \in \{2, 4, \ldots, 30\}$, $L \in \{1, 2, \ldots, 40\}$.

**Results**  Both CRP-LSH and ERP-LSH achieve a competitive efficiency over the other two methods. We list the test results in Table 3, and put parameters in Table 4 in Appendix C.

Table 3: Average query cost and average approximation rate if applicable (in parentheses).

|  | Corel_uci<br>($d = 32$) | Corel_hist<br>($d = 44$) | Cade<br>($d = 120$) | ImageNet<br>($d = 100$) |
|---|---|---|---|---|
| Linear scan | 61220 | 17809 | 11715 | 31458 |
| Metric tree | 2575 | 718 | 9184 | 12375 |
| CRP-LSH | $329 \pm 55 \ (1.07)$ | $245 \pm 43 \ (1.05)$ | $292 \pm 11 \ (1.11)$ | $548 \pm 66 \ (1.09)$ |
| ERP-LSH | $330 \pm 18 \ (1.11)$ | $250 \pm 15 \ (1.08)$ | $218 \pm 8 \ (1.15)$ | $346 \pm 15 \ (1.13)$ |

## 5  Conclusion

In this paper, we have proposed an explicit embedding from $\ell_1$ to $\ell_2^2$, and we have found an algorithm to generate the random projections, reducing the time dependence of $n$ from $O(n)$ to $O(\log n)$. In addition, we have observed that the effective rank of the (centered) embedding is as low as $O(d \ln n)$, compared to its rank $O(n)$. Algorithms remain to be explored, in order to take advantage of such a low rank.

Our current method takes space $O(ndm)$ to store the parameters of the random vectors, where $m$ is the number of hash functions. We have implemented one empirical scheme [39] to reuse the hashing functions. It is still expected to develop other possible schemes.

## Acknowledgement

The authors are grateful to the National Science Foundation for support under grant IIS-1162581.

## Footnotes

[2] Multiplying $H$ on the right side of any matrix will remove the mean for each column vector.

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

# A  Proof details

In this section, we give a proof of Theorem 1.

*Proof.* **Both joint distributions are multivariate Gaussian:** It is well-known that any linear combination of independent Gaussian random variables is Gaussian. In Eq. 7 and Eq. 8, the variables $[p_g(x_1), p_g(x_2), \ldots, p_g(x_n), p_g(q)]$ and $[p_e(x_1), p_e(x_2), \ldots, p_e(x_n)]$ are linear combinations of i.i.d Gaussian variables. Furthermore, according to Eq. 9, observe that $p_e(q)$ can be written as $p_e(q) = p_e(x_\alpha) + \frac{\sigma_1^2 \cdot (p_e(x_{\alpha+1}) - p_e(x_\alpha))}{\sigma_1^2 + \sigma_2^2} + Y$, where $Y \sim N\left(0, \frac{\sigma_1^2 \sigma_2^2}{\sigma_1^2 + \sigma_2^2}\right)$ is another independent variable. In all, the two joint distributions are multivariate Gaussian.

**Identical mean:** We prove that the means are identical. By linearity of expectation, and because $u$ and $v$ have zero mean, $\mathbb{E}[p_e(x_i)] = \mathbb{E}[p_g(x_i)] = 0$ for $i = 1, 2, \ldots, n$. Likewise, for a query $q$, we have $\mathbb{E}[p_g(q)] = 0$. It remains to determine $\mathbb{E}[p_e(q)]$:

$$
\begin{aligned}
\mathbb{E}[p_e(q)] &= \mathbb{E}_{u,\Delta}[p_e(x_\alpha)] + \mathbb{E}_{u,\Delta}[\Delta] \\
&= \mathbb{E}_u[p_e(x_\alpha)] + \mathbb{E}_u[\mathbb{E}_\Delta[\Delta | p_e(x_{\alpha+1}) - p_e(x_\alpha)]] \quad \text{(Tower Rule)} \\
&= 0 + \mathbb{E}_u\left[\frac{\sigma_1^2}{\sigma_1^2 + \sigma_2^2}(p_e(x_{\alpha+1}) - p_e(x_\alpha))\right] \\
&= \frac{q - x_\alpha}{x_{\alpha+1} - x_\alpha} \mathbb{E}_u[p_e(x_{\alpha+1}) - p_e(x_\alpha)] = 0
\end{aligned}
$$

**Identical covariance:** We prove the covariances are identical. For the ideal projection $p_g$, we claim that for any $x, y \in \mathcal{S} \cup \{q\} \subset \mathbb{R}$, $cov(p_g(x), p_g(y)) = \min\{x, y\} - x_1$. Here is the proof for $x_i, x_j \in \mathcal{S}$ where $x_i \le x_j \le q$ (the other cases are similar):

$$
\begin{aligned}
cov(p_g(x_i), p_g(x_j)) &= \mathbb{E}\left[\left(\sum_{k_1=1}^{i-1} v^{(k_1)}\sqrt{x_{k_1+1} - x_{k_1}}\right) \cdot \left(\sum_{k_2=1}^{j-1} v^{(k_2)}\sqrt{x_{k_2+1} - x_{k_2}}\right)\right] \\
&= \mathbb{E}\left[\sum_{k=1}^{i-1} \left(v^{(k)}\right)^2 \cdot (x_{k+1} - x_k)\right] = \sum_{k=1}^{i-1}(x_{k+1} - x_k) = x_i - x_1.
\end{aligned}
$$

For the actual projection that we use, $p_e$, there are four cases that need to be discussed. In the first case, consider $x_i \le x_j \in \mathcal{S}$ where $1 \le i \le j \le n$. The proof is similar to the above. We have $cov(p_e(x_i), p_e(x_j)) = x_i - x_1$.

Before we prove the other cases, we make some key observations. Note that for $p_e(q) = p_e(x_\alpha) + \Delta$ in Eq. 9, the random variable $\Delta$ only depends on the random variable $(p_e(x_{\alpha+1}) - p_e(x_\alpha)) = \sqrt{x_{\alpha+1} - x_\alpha} \cdot u^{(\alpha)}$, which implies that $\Delta$ is independent of $p_e(x_i)$ if $x_i \le x_\alpha$; $\Delta$ is also independent of $(p_e(x_i) - p_e(x_{\alpha+1}))$ if $x_i \ge x_{\alpha+1}$. In addition, $\mathbb{E}\left[(p_e(x_{\alpha+1}) - p_e(x_\alpha))^2\right] = \mathbb{E}\left[\left(u^{(\alpha)} \cdot \sqrt{x_{\alpha+1} - x_\alpha}\right)^2\right] = x_{\alpha+1} - x_\alpha = \sigma_1^2 + \sigma_2^2$.

Now, onto the remaining cases.

- Consider $x_i < q$ :

$$
\begin{aligned}
&cov(p_e(x_i), p_e(q)) \\
&= \mathbb{E}\left[p_e(x_i) \cdot (p_e(x_\alpha) + \Delta)\right] \\
&= \mathbb{E}\left[p_e(x_i) \cdot p_e(x_\alpha)\right] + \mathbb{E}\left[\mathbb{E}\left[p_e(x_i) \cdot \Delta | p_e(x_{\alpha+1}) - p_e(x_\alpha)\right]\right] \\
&= \mathbb{E}\left[p_e(x_i) \cdot p_e(x_\alpha)\right] + \mathbb{E}\left[\mathbb{E}\left[p_e(x_i) | p_e(x_{\alpha+1}) - p_e(x_\alpha)\right] \cdot \mathbb{E}\left[\Delta | p_e(x_{\alpha+1}) - p_e(x_\alpha)\right]\right] \\
&= \mathbb{E}\left[p_e(x_i) \cdot p_e(x_\alpha)\right] + \mathbb{E}\left[0 \cdot \mathbb{E}\left[\Delta | p_e(x_{\alpha+1}) - p_e(x_\alpha)\right]\right] \\
&= \mathbb{E}\left[p_e(x_i) \cdot p_e(x_\alpha)\right] = x_i - x_1
\end{aligned}
$$

- Consider $q$ and $q$:

$$cov(p_e(q), p_e(q))$$
$$= \mathbb{E}\left[(p_e(x_\alpha) + \Delta) \cdot (p_e(x_\alpha) + \Delta)\right]$$
$$= \mathbb{E}\left[p_e(x_\alpha) \cdot p_e(x_\alpha)\right] + 2\mathbb{E}\left[p_e(x_\alpha) \cdot \Delta\right] + \mathbb{E}\left[\Delta^2\right]$$
$$= x_\alpha - x_1 + \mathbb{E}\left[\Delta^2\right]$$
$$= x_\alpha - x_1 + \mathbb{E}\left[\mathbb{E}\left[\Delta^2 \middle| p_e(x_{\alpha+1}) - p_e(x_\alpha)\right]\right]$$
$$= x_\alpha - x_1 + \mathbb{E}\left[\frac{\sigma_1^2 \sigma_2^2}{\sigma_1^2 + \sigma_2^2} + \left(\frac{\sigma_1^2}{\sigma_1^2 + \sigma_2^2}(p_e(x_{\alpha+1}) - p_e(x_\alpha))\right)^2\right]$$
$$= x_\alpha - x_1 + \frac{\sigma_1^2 \sigma_2^2}{\sigma_1^2 + \sigma_2^2} + \mathbb{E}\left[\left(\frac{\sigma_1^2}{\sigma_1^2 + \sigma_2^2}(p_e(x_{\alpha+1}) - p_e(x_\alpha))\right)^2\right]$$
$$= x_\alpha - x_1 + \frac{\sigma_1^2 \sigma_2^2}{\sigma_1^2 + \sigma_2^2} + \frac{\sigma_1^4}{\sigma_1^2 + \sigma_2^2}$$
$$= x_\alpha - x_1 + \sigma_1^2 = q - x_1$$

- Consider $x_i > q$ :

$$cov(p_e(q), p_e(x_i))$$
$$= \mathbb{E}\left[p_e(x_\alpha) \cdot p_e(x_i)\right] + \mathbb{E}\left[\Delta \cdot p_e(x_i)\right]$$
$$= x_\alpha - x_1 + \mathbb{E}\left[\Delta \cdot (p_e(x_\alpha) + (p_e(x_{\alpha+1}) - p_e(x_\alpha)) + (p_e(x_i) - p_e(x_{\alpha+1})))\right]$$
$$= x_\alpha - x_1 + \mathbb{E}\left[\Delta \cdot (p_e(x_{\alpha+1}) - p_e(x_{\alpha-1}))\right]$$
$$= x_\alpha - x_1 + \mathbb{E}\left[(p_e(x_{\alpha+1}) - p_e(x_\alpha)) \cdot \mathbb{E}\left[\Delta | p_e(x_{\alpha+1}) - p_e(x_\alpha)\right]\right]$$
$$= x_\alpha - x_1 + \mathbb{E}\left[\frac{\sigma_1^2}{\sigma_1^2 + \sigma_2^2}(p_e(x_{\alpha+1}) - p_e(x_\alpha))^2\right]$$
$$= x_\alpha - x_1 + \sigma_1^2 = q - x_1$$

$\square$

## B    Rank and effective rank

In this section, we would like to show that generally the embedding vectors appear to have a nearly full rank $(O(n))$, but it turns out that the (centered) matrix always has a low effective rank $(O(d \log n))$ .

We introduce several notations. For a matrix $A \in \mathbb{R}^{m \times n}$, let $L = \min\{m, n\}$, and let $\sigma_1 \geq \sigma_2 \geq \cdots \geq \sigma_L \geq 0$ be singular values of $A$. The Frobenius norm is defined as

$$\|A\|_F = \sqrt{\sum_{i=1}^{m}\sum_{j=1}^{n}|A_{i,j}|^2} = \sqrt{\sum_{k=1}^{L}\sigma_k^2}. \tag{10}$$

The spectral norm is defined as

$$\|A\|_2 = \max_{\|x\|_2 \neq 0} \frac{\|Ax\|_2}{\|x\|_2} = \sigma_1. \tag{11}$$

The effective rank [32] is defined as

$$r_e(A) = \frac{\|A\|_F^2}{\|A\|_2^2} = \frac{\sigma_1^2 + \sigma_2^2 + \cdots + \sigma_L^2}{\sigma_1^2} \tag{12}$$

### B.1    1-dimension case

We first consider the 1-dimension data $x_1 \leq x_2 \leq \cdots \leq x_n \in \mathbb{R}$ (suppose $n$ is even) and their embedding $\phi(x_1), \phi(x_2), \ldots, \phi(x_n) \in \mathbb{R}^{n-1}$. Define the matrix

$$\Phi = (\phi(x_1), \phi(x_2), \ldots, \phi(x_n)) \in \mathbb{R}^{(n-1)\times n}. \tag{13}$$

We formulate the centered embedding matrix $A \in \mathbb{R}^{(n-1)\times n}$ that we will investigate in the rest of this subsection.

$$
\begin{aligned}
A &= \Phi \cdot H \\
&= (\phi(x_1) - \vec{\mu}, \phi(x_2) - \vec{\mu}, \ldots, \phi(x_n) - \vec{\mu}) = \\
&= \frac{1}{n} \times
\begin{pmatrix}
-(n-1)\sqrt{x_2 - x_1} & \sqrt{x_2 - x_1} & \cdots & \sqrt{x_2 - x_1} \\
-(n-2)\sqrt{x_3 - x_2} & -(n-2)\sqrt{x_3 - x_2} & \cdots & 2\sqrt{x_3 - x_2} \\
-(n-3)\sqrt{x_4 - x_3} & -(n-3)\sqrt{x_4 - x_3} & \cdots & 3\sqrt{x_4 - x_3} \\
\vdots & \vdots & \ddots & \vdots \\
-\sqrt{x_n - x_{n-1}} & -\sqrt{x_n - x_{n-1}} & \cdots & (n-1)\sqrt{x_n - x_{n-1}}
\end{pmatrix}
\end{aligned} \tag{14}
$$

where $H = I - \frac{1}{n}\vec{1}\cdot\vec{1}^{\tau} \in \mathbb{R}^{n\times n}$ [2], $I$ is the $n$-by-$n$ identity matrix, $\vec{1} \in \mathbb{R}^n$ is the all-1 vector, $\vec{\mu} \in \mathbb{R}^{n-1}$ is the mean of the embedding vectors $\vec{\mu} = \frac{1}{n}\sum_{i=1}^{n}\phi(x_i)$.

**Lemma 1** (Nearly full rank). *The matrix $A = \Phi H \in \mathbb{R}^{(n-1)\times n}$ in Eq. 14 has rank at least $(n-2)$, if $x_1, x_2, \ldots, x_n \in \mathbb{R}$ are all distinct.*

*Proof.* We observe that

- Rank($\Phi$) is $(n-1)$, if $x_1, x_2, \ldots, x_n$ are distinct, according to Eq. 1.

- Rank($H$) is also $(n-1)$. It has $(n-1)$ linearly independent eigenvectors $(\vec{e}_1 - \vec{e}_2), (\vec{e}_1 - \vec{e}_3), \ldots, (\vec{e}_1 - \vec{e}_n)$ for the eigenvalue 1, where $\vec{e}_i \in \mathbb{R}^n$, $i = 1, 2, \ldots, n$ are the standard bases; the rest eigenvector is $(1, 1, \ldots, 1)^{\tau}$ for 0. Therefore, $H$ is similar to a $(n-1)$ rank matrix, and $H$ also has rank $(n-1)$.

Then, we have $\text{Rank}(\Phi H) \geq \text{Rank}(\Phi) + \text{Rank}(H) - n \geq n - 2$. □

**Theorem 2** (Low effective rank). *Take $n$ to be even. The matrix $A = \Phi H \in \mathbb{R}^{(n-1)\times n}$ in Eq. 14 has effective rank at most $2(1 + \ln(n/2))$, for any $x_1, x_2, \ldots, x_n \in \mathbb{R}$.*

*Proof.* The main idea of the proof is that we construct an auxiliary matrix $B \in \mathbb{R}^{(n-1)\times n/2}$ such that

- $\|B\|_F^2 \geq \|A\|_F^2$

- $\|B\|_2^2 \leq 2\|A\|_2^2$

- $r_e(B) \leq 1 + \ln(n/2)$

Then, we have $r_e(A) \leq 2 \cdot r_e(B) \leq 2(1 + \ln(n/2))$.

This auxiliary matrix is

$$
B_{(n-1)\times n/2} =
\begin{pmatrix}
\sqrt{x_2 - x_1} & 0 & 0 & 0 \\
\sqrt{x_3 - x_2} & \sqrt{x_3 - x_2} & 0 & 0 \\
\sqrt{x_4 - x_3} & \sqrt{x_4 - x_3} & \sqrt{x_4 - x_3} & \vdots \\
\vdots & \vdots & \vdots & \ddots & 0 \\
\sqrt{x_{n/2+1} - x_{n/2}} & \sqrt{x_{n/2+1} - x_{n/2}} & \sqrt{x_{n/2+1} - x_{n/2}} & \cdots & \sqrt{x_{n/2+1} - x_{n/2}} \\
\vdots & \vdots & \vdots & \ddots & 0 \\
\sqrt{x_{n-2} - x_{n-3}} & \sqrt{x_{n-2} - x_{n-3}} & \sqrt{x_{n-2} - x_{n-3}} & \vdots \\
\sqrt{x_{n-1} - x_{n-2}} & \sqrt{x_{n-1} - x_{n-2}} & 0 & 0 \\
\sqrt{x_n - x_{n-1}} & 0 & 0 & 0
\end{pmatrix}
\tag{15}
$$

First, we prove that $\|B\|_F^2 \geq \|A\|_F^2$ by comparing the corresponding terms in Eq. 17 and Eq. 18 below.

$$\|A\|_F^2 = \frac{1}{n^2} \sum_{k=1}^{n-1} \left[ \left( k(n-k)^2 + (n-k)k^2 \right) (x_{k+1} - x_k) \right]$$

$$= \frac{1}{n} \sum_{k=1}^{n-1} k(n-k)(x_{k+1} - x_k) \tag{16}$$

$$= \sum_{k=1}^{n/2} k \cdot \frac{n-k}{n} (x_{k+1} - x_k) + \sum_{k=1}^{n/2-1} k \cdot \frac{n-k}{n} (x_{n-k+1} - x_{n-k}) \tag{17}$$

Take the sum of the squared entries row by row from $B$ :

$$\|B\|_F^2 = \sum_{k=1}^{n/2} k \cdot (x_{k+1} - x_k) + \sum_{k=1}^{n/2-1} k \cdot (x_{n-k+1} - x_{n-k}) \tag{18}$$

Second, we prove that $\|B\|_2^2 \leq 2\|A\|_2^2$. For any $y \in \mathbb{R}^{n/2}$, note that

$$By = A \cdot \begin{pmatrix} -y \\ rev(y) \end{pmatrix} \tag{19}$$

$$\|y\|_2^2 = \frac{1}{2} \left\| \begin{pmatrix} -y \\ rev(y) \end{pmatrix} \right\|_2^2 \tag{20}$$

where $rev(y) \in \mathbb{R}^{n/2}$, and $rev(y)_i = y_{n/2-i+1}$ for $i = 1, 2, \ldots, n/2$. For a 3-dimension example,

$$rev \begin{pmatrix} 1 \\ 2 \\ 3 \end{pmatrix} \implies \begin{pmatrix} 3 \\ 2 \\ 1 \end{pmatrix} \tag{21}$$

They implies that

$$\|B\|_2^2 = \max_{y \neq 0} \|By\|_2^2 / \|y\|_2^2 = 2 \max_{y \neq 0} \left\| A \begin{pmatrix} -y \\ rev(y) \end{pmatrix} \right\|_2^2 / \left\| \begin{pmatrix} -y \\ rev(y) \end{pmatrix} \right\|_2^2 \leq 2\|A\|_2^2 \tag{22}$$

We conclude that $\|B\|_2^2 \leq 2\|A\|_2^2$.

Finally, we prove that $r_e(B) \leq 1 + \ln(n/2)$. Take the sum of the squared entries column by column from $B$ :

$$\|B\|_F^2 = (x_n - x_1) + (x_{n-1} - x_2) + \cdots + (x_{n/2+1} - x_{n/2}) = \sum_{k=1}^{n/2} (x_{n+1-k} - x_k) \tag{23}$$

For any $y \in \mathbb{R}^{n/2}$,

$$
\begin{aligned}
\|By\|_2^2 &= \left[y_1^2 \cdot (x_2 - x_1)\right] + \left[(y_1 + y_2)^2 \cdot (x_3 - x_2)\right] + \cdots + \\
&\qquad \left[(y_1 + y_2 + \cdots + y_{n/2})^2 \cdot (x_{n/2+1} - x_{n/2})\right] \\
&\quad + \left[(y_1 + \cdots + y_{n/2-1})^2 \cdot (x_{n/2+2} - x_{n/2+1})\right] + \cdots + \\
&\qquad \left[(y_1 + y_2)^2 \cdot (x_{n-1} - x_{n-2})\right] + \left[y_1^2 \cdot (x_n - x_{n-1})\right] \\
&= y_1^2 \cdot \left[(x_n - x_{n-1}) + (x_{n-1} - x_{n-2}) + \cdots + (x_2 - x_1)\right] \\
&\quad + \left[y_2^2 + 2y_2 y_1\right] \cdot \left[(x_{n-1} - x_{n-2}) + (x_{n-2} - x_{n-3}) + \cdots + (x_3 - x_2)\right] \\
&\quad + \left[y_3^2 + 2y_3(y_1 + y_2)\right] \cdot \left[(x_{n-2} - x_{n-3}) + (x_{n-3} - x_{n-4}) + \cdots + (x_4 - x_3)\right] \\
&\quad + \cdots \\
&\quad + \left[y_k^2 + 2y_k(y_1 + \cdots + y_{k-1})\right] \cdot \\
&\qquad \left[(x_{n+1-k} - x_{n-k}) + (x_{n-k-1} - x_{n-k-2}) + \cdots + (x_{k+1} - x_k)\right] \\
&\quad + \cdots \\
&\quad + \left[y_{n/2}^2 + 2y_{n/2}(y_1 + \cdots + y_{n/2-1})\right] \cdot \left[(x_{n/2+1} - x_{n/2})\right]
\end{aligned}
$$

(24)

(25)

$$
= \sum_{k=1}^{n/2} \left[y_k^2 + 2y_k(y_1 + y_2 + \cdots + y_{k-1})\right]\left[x_{n+1-k} - x_k\right]
$$

(26)

We regroup Eq. 24 according to the largest index of $y$ in each item. For example, the largest y index for $y_k^2$ and $y_i \cdot y_k$ are both $k$, if $i \leq k$. Then, we have Eq. 25 and Eq. 26.

Choose $y_k = k^{-1/2}$, which implies

- $\|y\|_2^2 = \sum_{k=1}^{n/2} y_k^2 = \sum_{k=1}^{n/2} \dfrac{1}{k} \leq 1 + \ln(n/2)$

- In Eq. 26, the item $\left(y_k^2 + 2y_k(y_1 + \cdots + y_{k-1})\right) \geq 1$, for all $k = 1, 2, \ldots, n/2$. This is true for $k = 1$. For $k = 2, 3, \ldots, n/2$, note that $2y_k(y_1 + \cdots + y_{k-1}) \geq 2y_k \cdot (k-1) \cdot y_{k-1} = \dfrac{2\sqrt{k-1}}{\sqrt{k}} \geq 1$.

Thereby, for the above choice of $y$,

$$
\|By\|_2^2 \geq \sum_{k=1}^{n/2}[x_{n+1-k} - x_k] = \|B\|_F^2
$$

(27)

$$
\|B\|_2^2 \geq \|By\|_2^2/\|y\|_2^2 \geq \|B\|_F^2/(1 + \ln(n/2))
$$

(28)

We have the bound for $r_e(A)$

$$
r_e(A) \leq 2r_e(B) \leq 2(1 + \ln(n/2))
$$

(29)

$\square$

## B.2 Multidimensional case

In the multidimensional case, the centered embedding matrix $A$ is

$$
A = \begin{pmatrix} A^{(1)} \\ A^{(2)} \\ \vdots \\ A^{(d)} \end{pmatrix}
$$

(30)

where $A^{(i)}$ is the centered embedding matrix for dimension $i$, for $i = 1, 2, \ldots, d$. The columns of $A^{(i)}$ may not be arranged in the same order as in Eq. 14, but the norms $\|A^{(i)}\|_F$ and $\|A^{(i)}\|_2$ are unchanged by the ordering.

Table 4: The parameters we use for Cauchy LSH and embedding LSH.

|         |   | Corel_uci | Corel_hist | Cade | ImageNet |
|---------|---|-----------|------------|------|----------|
| CRP-LSH | R | 2.43      | 772079     | 4.06 | 2.31     |
|         | k | 18        | 18         | 16   | 12       |
|         | L | 24        | 38         | 28   | 24       |
| ERP-LSH | R | 0.99      | 547        | 1.78 | 1.36     |
|         | k | 14        | 12         | 22   | 18       |
|         | L | 32        | 35         | 29   | 29       |

Table 5: Perturbation on parameters for Cauchy LSH

|            | NN Distance | Parameter Set One | | | Parameter Set Two | | | Parameter Set Three | | |
|------------|-------------|--------|---|----|--------|----|----|---------|----|----|
|            |             | R      | k | L  | R      | k  | L  | R       | k  | L  |
| Corel_hish | 77208       | 386040 | 8 | 20 | 772079 | 18 | 38 | 1158118 | 24 | 40 |
| Cade       | 0.81        | 0.81   | 4 | 16 | 4.06   | 16 | 28 | 8.10    | 20 | 8  |

**Lemma 2** (Nearly full rank)**.** *The matrix $A \in \mathbb{R}^{d(n-1)\times n}$ in Eq. 30 has a rank at least $(n-2)$, if for some index $k$ where $1 \le k \le d$, $x_1^{(k)}, x_2^{(k)}, \ldots, x_n^{(k)} \in \mathbb{R}$ are all distinct.*

*Proof.* This is due to that $\mathrm{Rank}(A) \ge \mathrm{Rank}(A^{(k)})$ . $\square$

**Theorem 3** (Low effective rank)**.** *The matrix $A \in \mathbb{R}^{d(n-1)\times n}$ in Eq. 30 has an effective rank at most $2d(1 + \ln(n/2))$, for any $x_1, x_2, \ldots, x_n \in \mathbb{R}^d$.*

*Proof.* Without loss of generality, suppose $\|A^{(1)}\|_F^2 \ge \|A^{(i)}\|_F^2$ for $i = 1, 2, \ldots, d$. We observe that

- $\|A\|_F^2 \le d \cdot \|A^{(1)}\|_F^2$ .

- $\|A\|_2^2 \ge \|A^{(1)}\|_2^2$ . Note that for any $y \in \mathbb{R}^n$, $\|Ay\|_2^2 = \sum_{k=1}^d \|A^{(k)}y\|_2^2 \ge \|A^{(1)}y\|_2^2$.

Therefore,

$$r_e(A) \le d \cdot r_e(A^{(1)}) \le d \cdot 2(1 + \ln(n/2)) \tag{31}$$

$\square$

## C  Experimental parameters

We summarize the parameters that we use in Table 4.

In addition, we show that the parameters that we use are efficient for Cauchy LSH. We perturb the parameter $R$ in Table 5, with either smaller $R$ (Set One) or larger $R$ (Set Three). Then, we find good $k$ and $L$, given $R$. We show the test results in Table 6, which is performed on a randomly sampled 10% of points as queries. Each test is repeated 10 times.

Table 6: Test results for Cauchy LSH

| | Parameter Set One | | | Parameter Set Two | | | Parameter Set Three | | |
|---|---|---|---|---|---|---|---|---|---|
| | Time | Ratio | Approx. | Time | Ratio | Approx. | Time | Ratio | Approx. |
| Corel_hist | 290 | 90% | 1.05 | 263 | 91% | 1.05 | 340 | 93% | 1.04 |
| Cade | 476 | 90% | 1.1 | 306 | 92% | 1.1 | 576 | 89% | 1.1 |