[Reviews · NeurIPS 2016]

Reviewer 1

Summary

The paper shows an embedding from l_1 into l_2-squared, for the application of nearest neighbor search. The embedding method can be adapted for a new out-of-sample extension (i.e., a previously unseen query point). The authors also show experiments showing the competitiveness of their method.

Qualitative Assessment

While the fundamental technical contribution is interesting and should be published, the contextualization is very far from ideal. Here are some points that the authors should address: - besides the l_1 methods based on Cauchy distribution, there is also LSH methods directly for the l_1 space: just impose a randomly shifted grid (see, e.g., the description in the CACM article by Andoni & Indyk). - it is not clear than embedding l_1 into l_2^2 would be the best way to tackly the l_1 problem. In particular, if we want to solve a c-approximation under l_1, this would translate into requiring a \sqrt{c} approximation for l_2 after the embedding. I.e., the resulting problem in l_2 is harder! Luckily, in l_2, LSH achieves runtime n^{1/c^2} (cf, l_1 achieves runtime n^{1/c}); however the corresponding l_2 algorithns are much harder (compare the algorithms from the reference [10] versus [5]). Thus, it seems more beneficial to tackle l_1 problem directly. It is possible that, in practice, l_2 algorithms are much more performant, but this need to be argued well. The authors should mention also that an embedding from l_1 into l_2-squared is actually known from [Linial, London, Rabunovich, Combinatorica 1995]. That embedding proceeds by embedding l_1 into Hamming space using unary embedding, and then embedding Hamming into l_2-squared using identity mapping. Its disadvantage is that the "scale" needs to be known to achieve a bounded dimension (which is a common assumption in LSH in fact). The algorithm from this paper in fact can be seen as an efficient application of the JL dimension reduction on the embedding resulting from the LLR construction. This is what I find the most interesting contribution in this paper, and it should be played up. However, I cannot recommend a positive score due to the issues from above.

Confidence in this Review

3-Expert (read the paper in detail, know the area, quite certain of my opinion)


Reviewer 2

Summary

The paper presents a novel approach to l1-nearest neighbor search in which the data points are first 'lifted' to a high-dimensional space (dimension d x (n-1), where d is the original dimension and n the number of data points) so that the *squared* ell2-distance in the high-dimensional space equals the ell1-distance in the original space. In order to make this idea practical, random projections are applied to the data points after lifting, without explicitly mapping the points to the high-dimensional space. Altogether, all necessary operations can be performed with reasonable time and space complexities as given in Table 1. The performance of the proposed approach for (approximate) nearest neighbor search is finally evaluated on four data sets.

Qualitative Assessment

I feel that the paper could profit from having more theoretical results regarding the quality of the embedding even though it may seem obvious from the construction and the J-L lemma. A theoretical comparison (regarding quality of the embedding and time/storage complexities) to alternative approaches to the problem would be a worthwhile addition to me. I had some slight difficulties following the experimental evaluation. The performance metric 'average query cost' is not clear to me. Similarly, I do not understand what is meant 'overhead'. Also, the paper is not clear about how LSH and random projections are applied. My guess is that v * x is evaluated according to Algorithm 2, with v being Gaussian. Presentation-wise, I think it is not always necessary to split the one-dimensional / multi-dimensional case, as the multi-dimensional case always seems to boil down to coordinate-wise application of the one-dimensional case.

Confidence in this Review

2-Confident (read it all; understood it all reasonably well)


Reviewer 3

Summary

The authors present an embedding from l1-distances (squareroot) to l2-distance which facilitates fast nearest neighbor search.

Qualitative Assessment

The authors present a mapping from l1-distances (squareroot) to l2-distances, which can be extended to query data and, hence, can be employed for efficient nearest neighbor search. Generally speaking, the paper is well-written.

Confidence in this Review

1-Less confident (might not have understood significant parts)


Reviewer 4

Summary

Types of acceleration strategy about search with l2 distance have be widely researched. In some scenario such as measurements between probability distributions, the l1 distance is a natural choice. This paper finds explicit embeddings for instances where the l2 distance of embeddings is the same as (sqrt of) l1 distances in the original space. In addition, embedding of a query can also be computed at the test time with a random projection technique. This paper proves that the two joint distributions, the one projects both training instances and queries with random gaussian matrix and the one projects them separately with the algorithm proposed in the paper, are the same. Experiments on various datasets show the approximate rate and query cost of different methods, and the proposed methods can get better results.

Qualitative Assessment

With diverse uses of l1 distance, this paper proposes to find embeeddings where l2 distance between embeddings can approximate l2 distance in original space. Inspired by mature techniques in l2 distance retrieval, the algorithms in this paper can deals with high dimensional datasets efficiently. Strong points: 1. This paper is clear and well written. 2. Challenges of the l1 distance embedding and their solutions are described well. 3. Theoretical and empiricial results validate the effectiveness of this method. Some suggestions for the paper: 1. In theorem 1, equivalence between two joint distributions are proved. Whether the distribuition directly relates to the final retrieval performance? There needs some more explanations or theoretical results. 2. Diverse types of performance criteria can be evaluated to show the performance of the proposed method. 3. Time comparison about different implementations and real application results (such as image retrievial) can make the paper better to understand.

Confidence in this Review

2-Confident (read it all; understood it all reasonably well)


Reviewer 5

Summary

The paper presents a nearest neighbor (NN) search method for l1 distance. The key idea of the proposed method is to embed from l1 to l2 because NN search on l2 is studied well. The paper proposes an explicit embedding method to realize it. The paper is well organized. The idea of the proposed method is basically clearly written. On the other hand, I worry about evaluation. Since the proposed explicit embedding depends on the coordinates of points indexed and the given query, it requires sorting in each dimension (its impact can be n log n for indexing and log n for query time?). Due to this, it does not sound the proposed method is tractable in a really large-scale dataset. I am not sure if the query cost used as the evaluation criterion reflects such an important cost that can heavily affects to computational time for practical use.

Qualitative Assessment

Basically the paper is solid. One concern is if the definition of the query cost is reasonable. Query cost should reflect computational time.

Confidence in this Review

2-Confident (read it all; understood it all reasonably well)


Reviewer 6

Summary

The nearest neighbor (NN) problem asks for given n points in a vector space and an extra query point q, to find the nearest neighbor to q among these n points (with respect to a given metric). The most common metric used is the l_2 metric for which many algorithms are known which give approximate nearest neighbor. In this paper, the authors use a simple isometric embedding of (a set of points in) l_1 into (l_2)^2 in order to use the methods of NN from the Euclidean metric in the l_1 case. More precisely, a set of n points in a d dimensional space are isometrically embedded inside an O(dn)-(l_2)^2 space. The 2 main issues this method faces and their algorithm overcome are 1. While the space requirement to writing the embedding is O(n^2 d), the use of Gaussian projection method in the l_2 space allows the algorithm to "avoid" actually writing the embedding explicitly and hence use less memory space. 2. Once the n points are isometrically embedded, this embedding usually cannot be extended to another query point. Nevertheless, the algorithm can randomly choose the image in l_2 of the new query point so that the joint distribution (after the Gaussian projection) will remain the same.

Qualitative Assessment

The article is well written and mostly elementary. In particular, the embedding itself and the algorithms are very simple and easy to read. What is missing is a reference to algorithms based on approximate embedding, if there are such (it might be the last paragraph of section 1, lines 121-126, though I'm not sure). Other than that some few minor corrections: line 77 - "a\some Hilbert space" instead of "Hilbert space". line 204 - should be x_i and not q_i. line 215 - the projection is of the images of S,q and not S,q themselves. line 242 - "of the our method" should probably be "of our method".

Confidence in this Review

2-Confident (read it all; understood it all reasonably well)